# The Development of Solvent Cast Films or Electrospun Nanofiber Membranes Made from Blended Poly Vinyl Alcohol Materials with Different Degrees of Hydrolyzation for Optimal Hydrogel Dissolution and Sustained Release of Anti-Infective Silver Salts

**DOI:** 10.3390/nano11010084

**Published:** 2021-01-03

**Authors:** John Jackson, David Plackett, Eric Hsu, Dirk Lange, Robin Evans, Helen Burt

**Affiliations:** 1Faculty of Pharmaceutical Sciences, UBC, 2405 Wesbrook Mall, Vancouver, BC V6T 1Z3, Canada; davidplackett@gmail.com (D.P.); etjh111@gmail.com (E.H.); helen.burt@ubc.ca (H.B.); 2Stone Centre, Department of Urological Sciences, UBC, Vancouver General Hospital, Vancouver, BC V5Z 1M9, Canada; dirk.lange@stonecentre.ca; 3Ventura County Medical Centre, UCLA School of Medicine, Ventura, CA 93003, USA; robinjamesevans@gmail.com

**Keywords:** poly vinyl alcohol, electrospun membranes, anti-infective silver, wound dressing

## Abstract

Introduction: We previously described the manufacture and characterization of hydrogel forming, thin film, anti-infective wound dressings made from Poly Vinyl Alcohol (PVA) and silver nanoparticles, crosslinked by heat. However, these films were designed to be inexpensive for simple manufacture locally in Africa. In this new study, we have further developed PVA dressings by manufacturing films or electrospun membranes, made from blends of PVA with different degrees of hydrolyzation, that contain silver salts and degrade in a controlled manner to release silver in a sustained manner over 12 days. Methods: Films were solvent cast as films or electrospun into nanofibre membranes using blends of 99 and 88% hydrolyzed PVA, containing 1% *w/w* silver sulphadiazine, carbonate, sulphate, or acetate salts. Dissolution was measured as weight loss in water and silver release was measured using inductively coupled plasma (ICP) analysis. Results: Cast films generally stayed intact at PVA 99: PVA 88% ratios greater than 40:60 whereas electrospun membranes needed ratios greater than 10:90. Films (40:60 blend ratio) and membranes (10:90) all released silver salts in a sustained fashion but incompletely and to different extents. Electrospun membranes gave more linear release patterns in the 2–12 day period and all salts released well. Conclusion: Blended PVA cast films offer improved control over hydrogel dissolution and silver release without the need for high temperature crosslinking. Blended PVA electrospun membranes further improve membrane dissolution control and silver release profiles. These blended PVA films and membranes offer improved inexpensive systems for the manufacture of long lasting anti-infective hydrogel wound dressings.

## 1. Introduction

In our previous paper we described the development of inexpensive anti-infective wound or burn dressings for third world applications (accepted for publication in nanomaterials). The dressings were thin membranes manufactured from poly vinyl alcohol (PVA) containing silver as the anti-infective agent. 

However, although suitable for third world use because the dressings can be manufactured locally with minimal equipment, the dressings only have minor control of silver release profiles or hydrogel dissolution which might be a prohibitive challenge for use in modern hospital settings where clinicians want a more sophisticated product. A potential improvement to such a PVA based wound dressing might be the design of a dressing with a controlled dissolution profile, that did not require heating, that might be applied more easily, that released silver in a more controlled manner over longer time periods and did not use silver nitrate which can cause discoloration of skin [1].

PVA is ideally suited for use in wound dressing compositions because cast films may be applied as a slightly swollen hydrogel film to the wound where it stays in position but is essentially non adherent and may be washed off [2]. This feature keeps the wound moist for faster healing with less scarring and the material may then further absorb wound or burn exudate and form a physical barrier to prevent bacterial invasion [3,4,5,6]. Any residual film material is then easily removed without causing pain to the patient and damage to the wound healing area of skin.

PVA comes in a number of forms with variable molecular weights and variable degrees of hydrolyzation ranging from 80 to 99%. PVA is manufactured by the hydrolyzation of the acetate groups on poly vinyl acetate so that as the number of acetate groups falls the polymer chains become less soluble. In fact the solubility of PVA is inversely proportional to the degree of hydrolyzation whereby cast films made from 99% hydrolyzed PVA swell but dissolve poorly in water and 80 to 90% hydrolyzed PVA films swell but then dissolve rapidly [6]. It is generally accepted that some crosslinking of PVA is needed to create a robust, longer lasting wound dressing.

Crosslinking may involve the use of chemicals such as citric acid [7] or glutaraldehyde [2,8] but for wound dressing applications these methods need to ensure that all unused chemicals are removed before use. Other methods include ebeam [9] or gamma [4] irradiation which also serve to sterilize the films (perhaps not essential for an anti-infective containing dressings) and freeze thawing [4,10] which simply allows crystallites to form in the PVA which anchor polymer chains. In our previous study, we demonstrated that defined heat profiles combined with appropriate silver levels allowed for the effective cross linking of PVA membranes. However, the heating methods gave broad but not fine control of dissolution and silver release.

Other workers have blended PVA with other agents to try and produce membranes with improved performance characteristics. The agents include alginate [11,12], poly vinyl pyrrolidone [6], cellulose derivatives [7,11,13] collagen [14], polyethylene glycol (PEG) [10,15], chitosan [3], and latex [5]. Jodar et al. [2] created blends of PEG 4000 with PVA (98% hydrolyzed) and PVA (88% hydrolyzed) followed by glutaraldehyde crosslinking. These PEG blended films degraded over 1 to 24 h periods depending on composition allowing for some control of dissolution but almost all the silver sulphadiazine anti-infective agent was released within 100 min from these films. 

There is an unmet need for the development of PVA based wound dressings with improved performance so that the membranes may provide hydrogel protection but slowly dissolve releasing steady amounts of silver over the first few days. The objective of this study was to explore the use of blended PVA types with alternative forms of silver to achieve fine control of dissolution and silver release. This is important since all burns and wounds are different, requiring shorter or longer hydrogel protection and silver therapy depending on the severity or the trauma and the availability of clinical support. 

In this study we have found that PVA compositions manufactured from PVA types with different degrees of hydrolyzation (ranging from 80 to 99%) in various ratios but with no water soluble PEG added, allowed for controllable dissolution over many days. No heating was required and the inclusion of acetate, carbonate, sulphate or sulphadiazine salts of silver allowed for an effective antimicrobial composition. 

## 2. Materials and Methods

Poly(vinyl alcohol) (Selvol 540 88 mole% hydrolyzed, Mw ~ 150,000 and Selvol 125: 99 mole% hydrolyzed, Mw-125,000) was obtained from Sekisui Specialty Chemical Company, Dallas, TX, USA). Silver salts (>99.0%) were purchased from Sigma–Aldrich (St. Louis, MO, USA). All chemicals were used as supplied and without further purification. Deionized water was used in the preparation of all experimental PVA-silver formulations.

### 2.1. Methods

#### 2.1.1. Film Preparation (Solvent Cast PVA)

PVA was prepared as a 10% *w/w* stock solution by slowly adding PVA powder to a suitable volume of rapidly stirred water preheated to 85–90 °C followed by continued stirring and heating for approximately 60 min. When a clear solution had formed the vessel was removed from heating and cooled to room temperature. Stock silver salt solutions were prepared in water and stored covered with aluminum foil in a dark cupboard until required. Solutions of PVA were diluted down to 5% *w/w* and mixed together at the appropriate ratios. Finally, a small volume of the concentrated silver salt solution was then added in sufficient quantity to allow films to be cast in 60 mm × 15 mm disposable polystyrene Petri dishes to a final thickness of 100 um (Sarstedt Inc., Montreal, PQ, Canada). Generally, the % of silver ion (not the total weight of the salt) to PVA was 1%. The PVA-silver solutions in Petri dishes were loosely covered with aluminum foil and left in a 37 °C oven overnight, in order for the water to evaporate. All dried films were stored in a dark cupboard before evaluation.

#### 2.1.2. Manufacture of Electrospun Membranes

PVA electrospun membranes were manufactured using a Nanofibre Electrospinning Unit from Kato Tech Co. Ltd. (Kyoto, Japan using 10 mL of a 10% PVA polymer solution in water (no glycerol) containing silver salts where the ratio of the blend is described by the percentage of the 99% hydrolyzed PVA to the percentage of the 88% hydrolyzed PVA. Films were electrospun overnight (30 KV, 15 cm range, 0.1 mm/min syringe flow rate) and collected onto aluminum foil and stored at room temperature in the dark.

#### 2.1.3. Film Swelling and Dissolution Studies

PVA films containing silver and with a thicknesses of 100 μm (60 mg) were prepared as described above. These films were then stored for one week in the dark before use. Small sections of films (approximate diameters of 2 cm) were then placed on moist 0.2 μm filter discs (Millipore, Billerica, MA, USA) and weighed. The films and filters were covered with a thin layer of deionized water and left for appropriate times. After set time points the filter discs and adherent PVA-silver gel were moved to a Millipore vacuum apparatus and all excess water was removed from the filter over approximately 15 s. The combined PVA gel and filter were reweighed and recovered with a fresh layer of excess water. The weight gain (swelling) and weight loss (dissolution) were then calculated as a percentage of the original dry film weight.

#### 2.1.4. Silver Release Studies and Characterization

Films or electrospun membranes (10 mg) containing silver were placed in deionized water (5 mL) and the water fully removed at regular intervals for silver analysis by Inductively Coupled Plasma analysis. 5 mL of fresh water was then added to the films or membranes. Silver calibration standards were run every 10 samples. The instrument held reproducible standard curves over 100 sequential silver analysis with detection limits approximating 10 ng/mL. Each release study was run in triplicate for at least two weeks and the results plotted as the calculated percent silver released as a function of time. 

#### 2.1.5. Antibacterial Activity of Silver-Loaded PVA Films

In order to evaluate the antibacterial activity of PVA-silver formulations, all cast films and electrospun membranes were weighed at 20 ± 0.5 mg. The strain of *Staphylococcus aureus* (*S. aureus* Xen36 derived from ATCC 49525 (Wright)). was cultured overnight from freezer stocks (20 μL bacterial sample into 20 mL Luria Bertani (LB) broth), followed by a second sub-culture (200 μL in 20 mL LB broth) until an optical density of ~0.3–0.5 at *λ* = 600 nm was reached (measured using an Eppendorf BioPhotometer (Eppendorf, Mississauga, ON, Canada), at which point bacteria were used as outlined below.

The antibacterial activity of PVA-silver films was assessed by placing pre-weighed (20 mg) film or electrospun membrane samples containing 1% *w/w* silver loadings of either silver acetate, sulphadiazine, carbonate or sulphate in nutrient media containing *S. aureus* (5.00E + 05 colony forming units (CFU)/mL) with 10 mL of 100% culture media per bottle. The sample bottles were incubated at 37 °C with shaking at 100 rpm for 48 h. Aliquots were taken at 0, 6, 24, and 48 h following inoculation and bacterial numbers were assessed via CFU counts of serial dilutions of the aliquots on LB Agar following incubation in a 37 °C incubator for 12–16 h post-plating. 

## 3. Results

### 3.1. Film Casting

The use of 5% PVA solutions PVA polymers in water containing 2.5% glycerol was found to provide the optimal method of manufacture so that films were uniformly thin (approximately 100 um), flexible without cracking. Generally, films were a slightly whitish semi opaque colour with the silver carbonate films being slightly brown (which was the colour of the salt itself). Silver sulphadiazine is poorly soluble in water so these films had observable small particles embedded in the films but with a homogeneous dispersion still present in the final dried film. 

### 3.2. Dissolution of PVA Cast Films

For these studies, blends of 99% hydrolyzed and 88% hydrolyzed PVA were used as these are the most common and inexpensive forms of PVA available around the world. Initially films were cast without silver salts to assess general blend dissolution properties. It was observed that films containing 50% or more of the PVA 99% type did not fully dissolve over a week but those containing just 30% did dissolve. Therefore, a range of PVA blends (99:88) of 32:68 to 46:54 was used in these studies. Generally, for all four salts, the films swelled rapidly to approximately 400% and declined to less than 150% within 2 h. After that, swelling levels tended to stabilize and drop only slowly over the next few hours. Data are shown for cast films containing silver sulfadiazine, silver carbonate, silver sulphate and silver acetate in Figure 1, Figure 2, Figure 3 and Figure 4. Furthermore, films containing PVA 99 contents greater than 40% did not fully degrade and swelling remained above 0%. 

Note that a 0% swelling means the weight of the film is the same as the dry weight so some material has been lost as some water is present in the remaining film. To fully dissolve a value of −100% weight must be attained and none of the films were fully dissolved after 13 days. However, films at −50% weight were no longer intact and very fragmented. 

The swelling studies were followed for over ten days with weights being taken at 1, 2, 6, and 13 days. Most of the values remained the same as at 6 h so this data is not shown so as not to compact the x axis graph. 

There were some noticeable differences in swelling between PVA films containing different salts. Films with either the acetate or sulphadiazine salts were more robust and less soluble at 1 h with lower PVA 99% and higher PVA 88% content than films made with sulphate or carbonate salts. For example, silver acetate containing films with PVA 99% contents of 44, 42, and 40% all stayed at or close to 0% swollen at 6 h (and through to 13 days data not shown), whereas, for the sulphate salt, only 48 and 46% PVA 99% content films remained above the 0% swelling point. 

Overall these data show that fine control of film dissolution may be achieved for all four salts by changing the blending % of PVA (99:88% hydrolyzed) to have a PVA (99% hydrolyzed) content in the range of 30 to 50% by weight to PVA 88% hydrolyzed. 

### 3.3. Silver Release Studies

The release of silver salts from solvent cast films is shown in Figure 5. Films containing 60% content of PVA 88% were used in these studies because the films did not fully dissolve over extended periods in water and the 60% (PVA 88%) content represented a mid-point in the degradation studies. Films (10 mg) containing silver were placed in deionized water (5 mL) and the water fully removed at regular intervals for silver analysis by Inductively Coupled Plasma analysis. Five mL of fresh water was then added to the films. The order of levels of drug release were carbonate > sulphate > acetate > sulphadiazine.

### 3.4. Membrane Electrospinning

Using the described electrospinning methods, PVA blends containing 1% silver salts produced strong, tissue-like, thin membranes composed of nanofibers with a diameter of approximately 700 nm. The type of silver salt had little impact on the characteristics of the membranes, except that the silver carbonate membranes were slightly brown (like the cast films). These tissue-like membranes were robust and handleable with a strength lying between plastic wrap used in kitchens and tissue paper. The membranes were not statically self-adhering but were thin and very flexible and it was not advisable to squeeze them into a ball or they were difficult to smooth out again. Because of the inherent flexibility, glycerol was not included in the preparation. Overall, if packaged on a backing paper (or made thicker) these membranes could be easily applied or stretched over a wound. 

### 3.5. Dissolution of Electrospun PVA Membranes

Initial studies using electrospun membranes without silver, revealed that (compared to cast films) much higher levels of PVA 88% could be included in the PVA membranes without swelling or dissolution levels dropping to 0% or lower. Therefore, silver salt loaded membranes were electrospun using PVA (99%) content of 50 to 0% (i.e., with PVA (88%) of 50 to 100%). 

Generally, electrospun membranes composed of blends of PVA (99%) and PVA (88%) swelled more than the solvent cast films. All compositions (except those containing 100% *w/w* PVA (88%)) initially swelled to between 500 and 800% with only minor and slow reductions in swelling over the next 6 h (Figure 6, Figure 7, Figure 8 and Figure 9). For all salts, membranes containing PVA (99%) contents of 20% or more remained swollen at levels over 100% for more than 11 days (data points at 1, 2, 4, 7, and 11 days) not shown to avoid x axis compaction as graphs data points were largely unchanged.

### 3.6. Silver Release Studies

The release of silver salts from electrospun membranes is shown in Figure 10. Membranes containing 90% content of PVA 88% were used in these studies because the films did not fully dissolve over extended periods in water yet gave early signs of controlled dissolution. Electrospun membranes (10 mg) containing silver were placed in deionized water (5 mL) and the water was fully removed at regular intervals for silver analysis by Inductively Coupled Plasma analysis. 5 mL of fresh water was then added to the membranes. The profile of release was similar to that observed for the solvent cast films. There was a burst phase of release of between approximately 15 and 35% over the first 4 h, followed by a steady almost linear phase of release to day 3 (27 to 75%), followed by a slower prolonged release to day 11. As for cast films, the carbonate-containing membrane released the most drug with sulphate followed by acetate release rates below that. Opposite to films, electrospun membranes containing silver sulphadiazine released silver very rapidly (with the same kinetics as silver from carbonate membranes). For carbonate and sulphadiazine membranes, almost 100% silver release was achieved by day 11, whereas less silver was released for sulphate (75%) or acetate (45%) at the same time. 

### 3.7. Antibacterial Activity

All cast films and electrospun membranes were bactericidal (100% bacterial death) as shown in Figure 11. The antibacterial activity was common to all silver sulphadizine, carbonate, sulphate, and acetate salts. PVA films containing no silver had no effect on the rapid growth of bacteria. 

## 4. Discussion

There are two types of dressing for the treatment of wounds and burns composed of either dry or wet materials. These dressings may or may not contain anti-infective agents. The principle of using a dressing is to present a barrier to further microbial invasion, to soak up wound exudates and to possibly deliver an anti-infective agent, most commonly silver. Dry coverings like Acticoat™ containing silver nanoparticles are attractive as they are easy to handle and may be simply applied by taping to a skin. Furthermore, although silver generally is well regarded as an anti-bacterial agent, silver nanoparticles are now reported to be more effective and are regarded as the gold standard form of silver for wound dressings [4,15,16,17,18]. However, these dressings are very expensive especially for third world applications and dry dressings suffer from patient morbidity problems of pain and interference with healing when removed. Wet dressings are reported to offer improved wound healing over dry [6]. This is largely because the moist surface is more comfortable on the skin and removal may be facilitated by rinsing off with water without damage to the wound. 

A quick review of silver containing wound dressings on the internet reveals hundreds of products in the USA ranging for low cost dry dressings like Tegaderm (3 M) at less than $1 a dressing to more expensive Acticoat (Smith and Nephew, Watford, UK) at approximately $10 per unit. Wet dressings like alginate hydrogel dressings with silver (such as Algicell from Derma Sciences, Princeton, NJ, USA) cost approximately $9 each. In Africa, where fires burn inside homes, for cooking etc., there are many children who fall into such fires and incur serious and large skin burns requiring large areas of dressing material. Therefore, what is needed are inexpensive dressings that can preferable be manufactured without technical expertise. In third world countries, these dressings would preferably offer the best features of dressings sold in first world countries (moist hydrogels containing sustained release silver) but be inexpensive and manufactured locally. In the previous paper (accepted for publication in nanomaterials), we described the manufacture of such a dressing that used very inexpensive materials and could be manufactured in any facility with a cooking oven. Specific heating cycles and silver contents allowed for the crosslinking of the PVA films. However the dissolution rate of the films and the release of silver, although acceptable, was uncontrolled and the cast films required the addition of glycerol for flexibility. 

In this study we attempted to improve on the previously described dressing by producing films prepared without heat cycles and no silver assisted crosslinking where the dissolution and release of silver was controlled by the blended nature of the PVA components making up the film and by the choice of silver salt used in the dressing. We initially observed that cast films made from PVA with different degrees of hydrolyzation broke up or dissolved at different rates so that films made from pure 80, 88, 92, and even 94% hydrolyzed PVA dissolved in water by 4 h, with 94% being the slowest to dissolve. Films made from 96% hydrolyzed PVA dissolved over more than one day, and those with higher degrees of hydrolyzation were essentially insoluble (data not shown). The inverse relationship between the rate of PVA dissolution and the % degree of hydrolyzation of PVA has been previously reported [6]. We then experimented with blending PVAs with different PVA content to observe whether dissolution or degradation rates could be finely controlled. Using the PVAs with 99 and 88% hydrolyzation levels (the most commonly available), it was quickly determined that films containing more than 50% *w/w* of the 99% PVA swelled in water but did not fully degrade over 2 weeks whereas films with 36% or less PVA 99% content degraded quickly over a few hours (Figure 1, Figure 2, Figure 3 and Figure 4). There were only minor differences in the dissolution rates of films containing different silver salts with sulphadiazine and acetate containing films requiring less of the insoluble PVA 99% content than carbonate or sulphate films to stay intact over longer periods. 

Other workers have blended or modified PVA film composition to affect dissolution rates but these methods centre on ways to inhibit any breakdown of the PVA. Crosslinking with citric acid, glutaraldehyde, radiation, or freeze thawing have all been described for this purpose [2,4,7,8,9,10]. Other workers have blended a wide range of other agents in the PVA films such as polyvinylpyrollidone, cellulosic materials, chitosan or latex to provide long lasting films [3,5,7,9,11]. Interestingly, Jodar et al. [2] described blends of 99% hydrolyzed PVA with 88% hydrolyzed PVA which included the water soluble excipient polyethylene glycol and the drug silver sulphadiazine (1%) to produce long lasting films with good antimicrobial activity. However, in that study, the films were also crosslinked with glutaraldehyde and surprisingly all the silver salt eluted within 100 min, rendering the hydrogel long lasting for wound coverage, but potentially containing residual glutaraldehyde with no sustained silver release. In this study, using PVA 99%:PVA 88% (40:60) films, sustained silver release over 2 weeks was established for all four silver salt thus providing gently degrading films with a prolonged antimicrobial profile. 

With a well-established scientific base supporting the use of PVA in wound dressings, most new research has focused on the manufacture of electrospun PVA membranes composed of a network of polymeric nanofibers [8,19,20,21]. These nanofiber membranes may provide a more flexible membrane with interesting dissolution and drug release properties. As with PVA films, most workers focused on reducing membrane dissolution in water by crosslinking with glutaraldehyde [8], boron [19], methanol [21], polyacrylic acid plus heat [17]. Electrospun membranes containing silver demonstrated good antibacterial (against both gram positive and negative) activity [8,22]. In this study we demonstrated the effective manufacture of blended PVA nanofiber membranes. These nanofibers were approximately 500–800 nm in diameter similar to other workers [19,20] and formed on a revolving metal drum as a membrane (sheet-like: approximately 30 cm × 20 cm) to whatever thickness was desired. The membranes were very strong so that even ultrathin materials resisted tearing and self-adherence (Figure 12).

Most blended electrospun membranes stayed intact and swollen in water even with up to 90% (*w/w*) of PVA 88% content (Figure 6, Figure 7, Figure 8 and Figure 9), unlike solvent cast films which required approximately 40% *w/w* of PVA 99% content to avoid rapid disintegration. Whereas cast films swelled but then lost weight rapidly before stabilizing, electrospun membranes remained swollen over 12 days. This persistent swelling with only slow weight loss occurred without any of the crosslinking methods used by others. Jannesari et al. [23] reported that antibiotic (ciprofloxacin) loaded PVA nanofibres swelled by 800% and lost only 40% of weight after 24 h in water and that drug release continued for 3 days. It seems quite clear that electrospun PVA blends containing silver are more resistant to water dissolution than cast films and that crosslinking methods used by others are not necessary with electrospun membranes unless zero weight loss is needed. These membranes also gave excellent release profiles for 12 days for all silver salts (Figure 10). These profiles included a fast release over 3 days (to potentially eliminate existing bacteria in a wound) followed by an almost linear release over the next 10 days (to maintain a high silver concentration to kill invasive bacteria during wound healing). Similarly, Zhang [22] and Polivkova [18] both showed good antibacterial activity of silver against gram positive staphylococcus bacteria when released from silver nanostructures. 

All silver containing cast films and electrospun membranes were 100% bactericidal against drug resistant Staph Aureus, clearly demonstrating the potential ability of all the dressings irrespective of silver salt type to eliminate all existing bacteria in a wound (Figure 11). This is the most important feature of an anti-infective dressing. 

Clearly solvent casting of silver loaded blended PVA films presents as the simplest method for manufacturing antibacterial wound dressings in the third world. Only a simple heating cycle is needed and the addition of a small amount of glycerol renders the films very flexible. All materials (including silver salts) are inexpensive and silver salts are only needed at 1% *w/w* loadings to function as antibacterial materials. However, although the release profiles are good for carbonate and sulphate salts, lower levels of release were observed from day 2 to 12 for the acetate and sulphadiazine salts and this may not offer an optimal profile for these salts. Additionally, the dissolution profiles show that a small error in the PVA 99%: PVA 88% recipe could result in big changes in weight loss and residence time on a wound. Because the films are stiff they may be better applied moistened which might require a medically minded person to apply clean water to the wound for application of the film by gentle holding in place to swell and adhere. Another aspect of PVA cast films is that they need to be moderately thick (e.g., 100 um) to handle so that upon initial swelling in excess moisture environments, these films may become quite heavy and potentially mobile unless covered by a secondary dressing. 

On the other hand, electrospun blended PVA membranes offer the same ability to control dissolution profiles as cast films but they have the clear advantage of potentially being a lightweight, easy-to-apply membrane which have a large capacity to absorb exudate over a long period of time without significant weight loss. Furthermore, the release profiles of silver for all four salts demonstrated near perfect sustained release for nearly two weeks by which time the wound may have almost healed with little need for the dressing. Although the electrospun membranes in this study were produced in an expensive commercial machine, a simple inexpensive set up can be easily assembled using cheap materials. Initially, in this lab an empty food can was connected to a slow speed electric drill to act as the revolving collection surface, a needle or syringe system was set up with a cheap syringe pump and the can and the needle tip were connected to an inexpensive high voltage generator approximately 20 cm apart. This system produced excellent electrospun membranes just like the commercial machine. 

## 5. Conclusions

In this study, novel methods of manufacturing anti-infective, PVA hydrogel dressings in the form of cast films or electrospun nanofiber membranes are described. By varying the blending ratios of PVA polymers with high (insoluble) and low (soluble) degrees of hydrolyzation, films with varying degrees and rates of dissolution in aqueous environments were prepared. Both films and membranes released silver ions in a controlled manner with steady sustained release over four days, which is a time frame likely to exceed the preferred residence time on a wound. These dressings were found to be easy to apply, to hydrate into a hydrogel rapidly and by varying the PVA ratios may provide a hydrogel protective covering with tunable dissolution and silver release profiles for burns and wound settings. All films and membranes were bactericidal against gram positive staphylococcus bacteria. These low cost films and membranes may be suitable for treating burns and wounds in all world settings.

## Figures and Tables

**Figure 1 nanomaterials-11-00084-f001:**
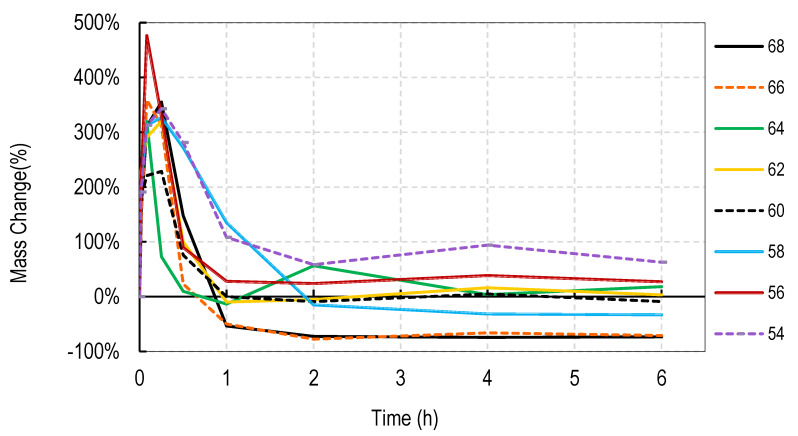
Degradation profile of blended PVA cast films (99%:88% hydrolyzed) containing 1% *w/w* Silver Sulphadiazine shown as *w/w*% content of PVA 88% hydrolyzed in the film.

**Figure 2 nanomaterials-11-00084-f002:**
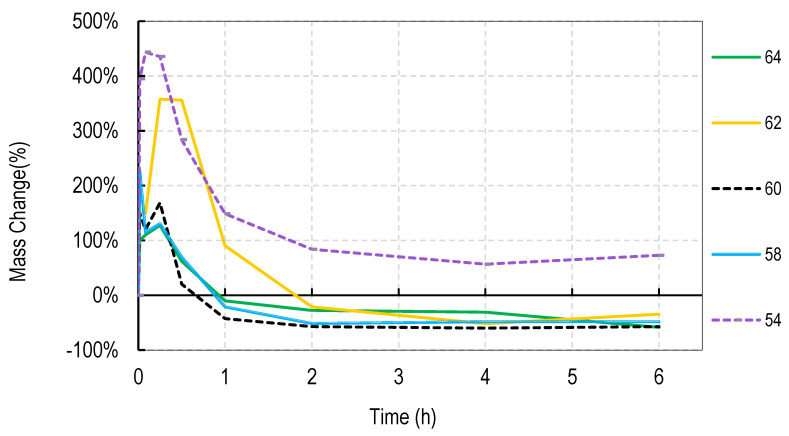
Degradation profile of blended PVA cast films (99%:88% hydrolyzed) containing 1% *w/w* Silver Carbonate shown as *w/w*% content of PVA 88% hydrolyzed in the film.

**Figure 3 nanomaterials-11-00084-f003:**
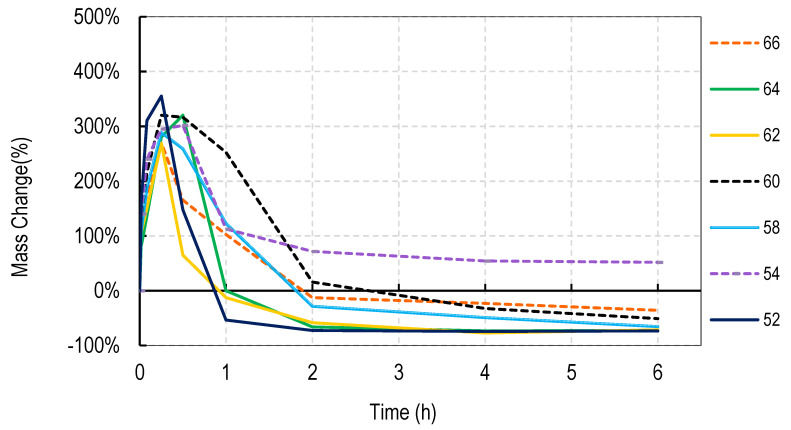
Degradation profile of blended PVA cast films (99%:88% hydrolyzed) containing 1% *w/w* Silver Sulphate shown as *w/w*% content of PVA 88% hydrolyzed in the film.

**Figure 4 nanomaterials-11-00084-f004:**
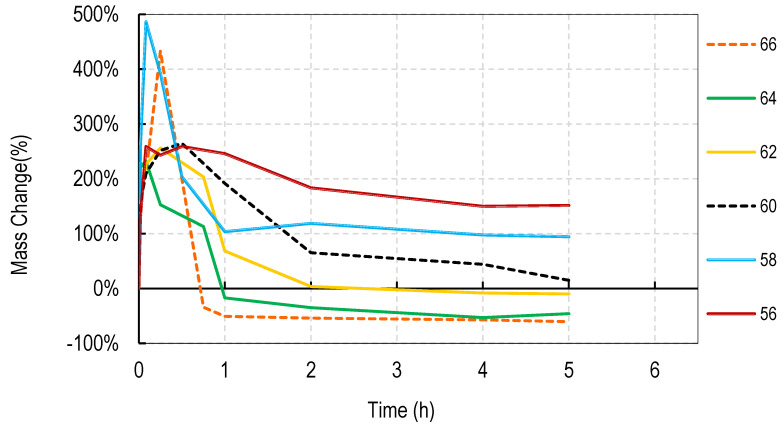
Degradation profile of blended PVA cast films (99%:88% hydrolyzed) containing 1% *w/w* Silver Acetate shown as *w/w*% content of PVA 88% hydrolyzed in the film.

**Figure 5 nanomaterials-11-00084-f005:**
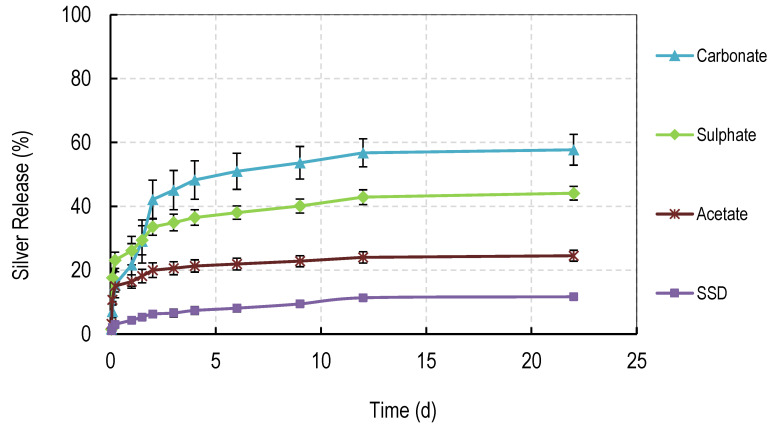
Release of Silver from PVA cast films made from 40:60 (weight ratio) of PVA 99%:PVA 88% hydrolyzed loaded with 1% *w/w* of various silver salts.

**Figure 6 nanomaterials-11-00084-f006:**
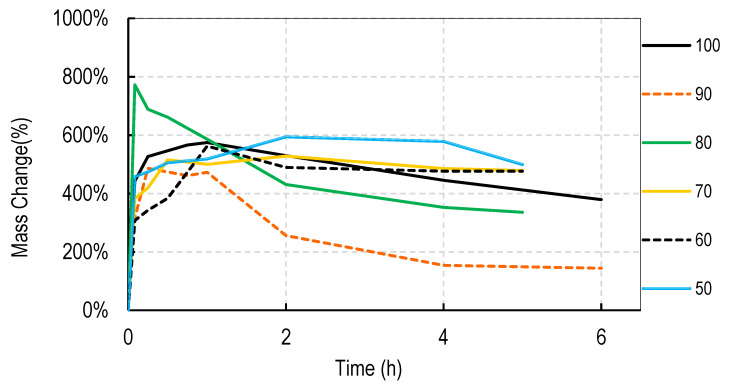
Degradation profile of blended electrospun PVA membranes (99%:88% hydrolyzed) containing 1% *w/w* Silver Sulphadiazine shown as *w/w*% content of PVA 88% hydrolyzed in the membrane.

**Figure 7 nanomaterials-11-00084-f007:**
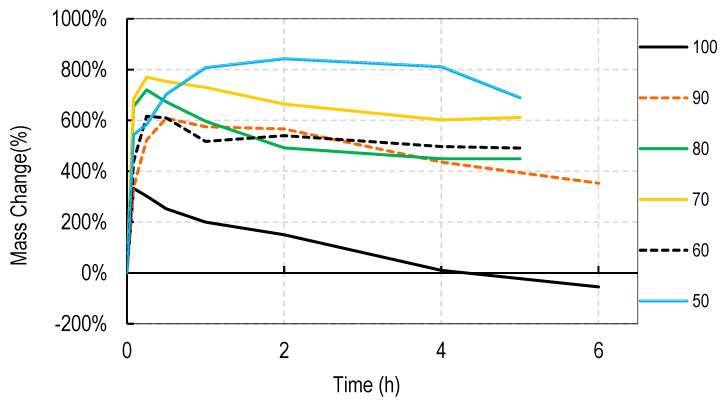
Degradation profile of blended electrospun PVA membranes (99%:88% hydrolyzed) containing 1% *w/w* Silver Carbonate shown as *w/w*% content of PVA 88% hydrolyzed in the membrane.

**Figure 8 nanomaterials-11-00084-f008:**
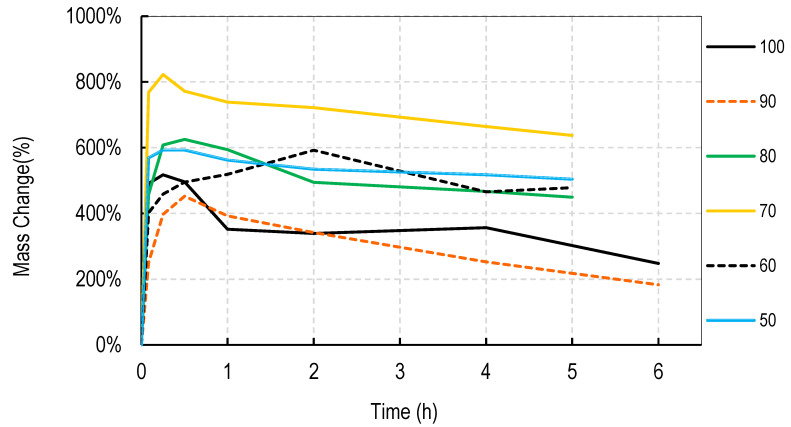
Degradation profile of blended electrospun PVA membranes (99%:88% hydrolyzed) containing 1% *w/w* Silver Sulphate shown as *w/w*% content of PVA 88% hydrolyzed in the membrane.

**Figure 9 nanomaterials-11-00084-f009:**
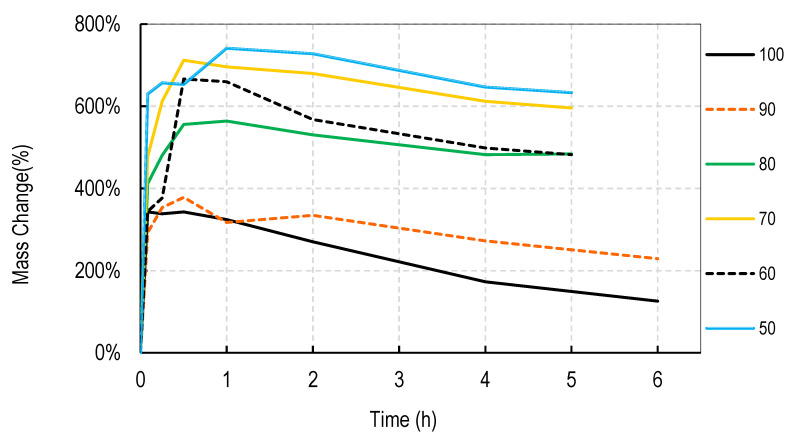
Degradation profile of blended electrospun PVA membranes (99%:88% hydrolyzed) containing 1% *w/w* Silver Acetate shown as *w/w*% content of PVA 88% hydrolyzed in the membrane.

**Figure 10 nanomaterials-11-00084-f010:**
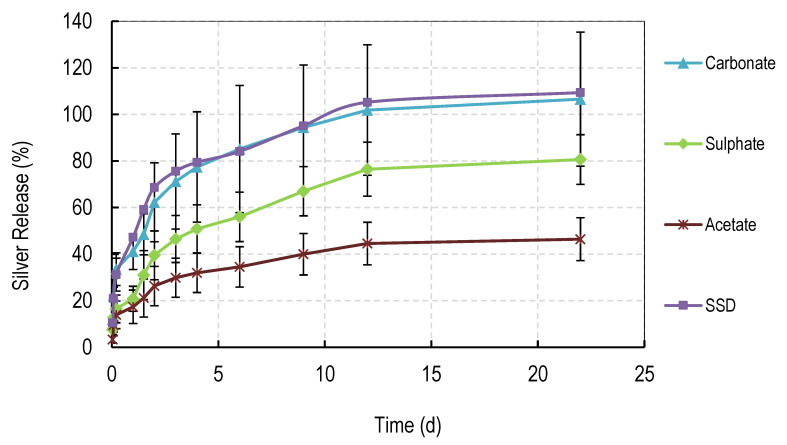
Release of Silver from PVA electrospun membranes made from 10:90 (weight ratio) of PVA 99%:PVA 88% hydrolyzed loaded with 1% *w/w* of various silver salts.

**Figure 11 nanomaterials-11-00084-f011:**
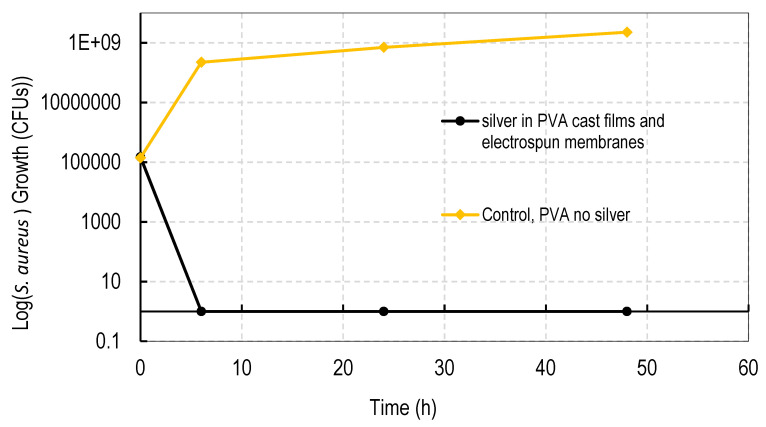
Anti-bacterial (*S. aureus)* activity of silver loaded PVA films made from 40:60 (weight ratio) of PVA 99%:PVA 88% hydrolyzed loaded with 1% *w/w* of various silver salts, or, electropsun membranes made from 10:90 (weight ratio) of PVA 99:PVA 88% hydrolyzed loaded with 1% *w/w* of various silver salts. All four films and four membranes (for all four salts: silver sulphadiazine, carbonate, sulphate, and acetate) killed all MRSA bacteria whereas the PVA control film (no silver) had no effect on bacterial growth.

**Figure 12 nanomaterials-11-00084-f012:**
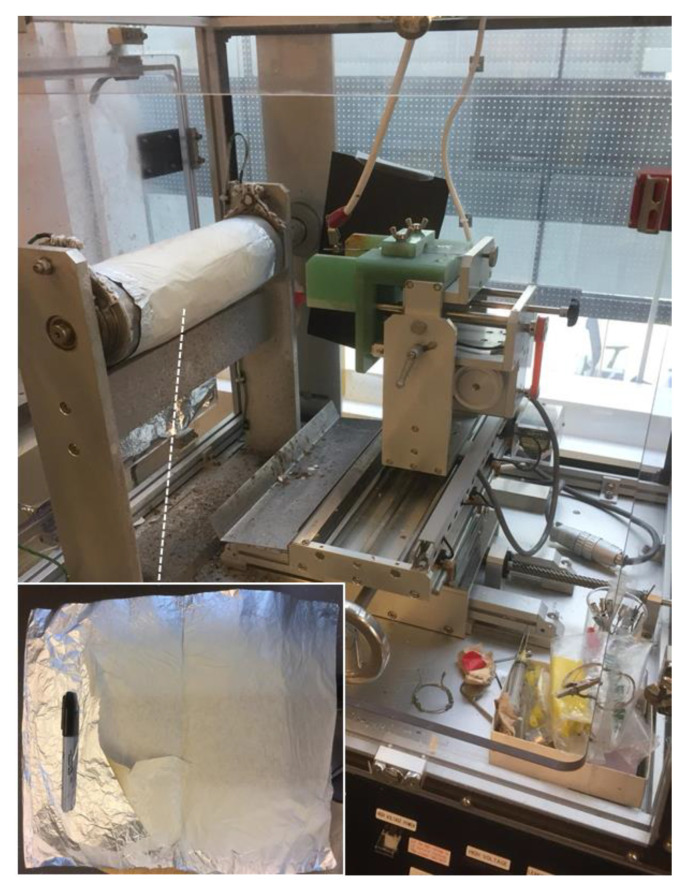
Electrospinning apparatus and membranes. Shows the nanofiber electrospinning unit (Kato) and a electrospun PVA membrane (10% PVA (99%):90% PVA (88%)) containing silver carbonate (1%).

## Data Availability

MDPI is committed to supporting open scientific exchange and enabling our authors to achieve best practices in sharing and archiving research data. We encourage all authors of articles published in MDPI journals to share their research data. More details in section “MDPI Research Data Policies” at https://www.mdpi.com/ethics.

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
