# Peer review of "The Development of Solvent Cast Films or Electrospun Nanofiber Membranes Made from Blended Poly Vinyl Alcohol Materials with Different Degrees of Hydrolyzation for Optimal Hydrogel Dissolution and Sustained Release of Anti-Infective Silver Salts"

_nanomaterials, 2021, doi:10.3390/nano11010084_

Round 1

Reviewer 1 Report

In this manuscript, the authors reported an interesting work which developed silver salts loaded PVA blended films/membranes by solvent casting or electrospinning, and studied the effects of PVA types/hydrolyzation and silver salts on the PVA dissolution and silver release. This work provided a cost-effective approach without crosslinking of PVA to achieve tunable and controlled silver release. However, some of the important data such as membrane morphology is missing, and the manuscript is not well written. Therefore, it is not recommended for publication before major revision. Below please find the comments/questions.

(1) The background information about would dressing, e.g., research progress, challenges, etc.

(2) In the Introduction, paragraph 4, please explain why PVA solubility is inversely proportional to the degree of hydrolyzation. What the mechanism?

(3) In the last paragraph in the Introduction, please indicate what the alternative salts of silver are.

(4) SEM images and EDX data should be provided to show the morphology of resulting membranes (film casting and electrospinning) and prove the homogenous dispersion of Silver salt.

(5) Degradation is not a proper term to describe the PVA films weight loss. PVA can not be degraded into small molecules in such a short time. It is the "dissolution" of PVA which leads to the weight loss. Please revise it accordingly in the whole manuscript.

(5) I would suggest to combine Figure 1-4 into one figure, and also combine Figure 6-9.

(6) SEM images of electrospun PVA/silver salts membranes before and after dissolution should be provided to show the morphology change.

(7) The order for Figure 5 is not correct.

(8) Why 40:60 (PVA ratio) for cast film was selected to do the release kinetics study?

(9) Why 10:90 (PVA ratio) for electrospun membrane was selected to do the release kinetics study?

(10) It’s better to move the first two paragraphs from the Discussion to the Introduction.

(11) Can not find Figure 12 in the manuscript.

(12) Conclusion section is missing.

Reviewer 2 Report

The manuscript "The development of solvent cast films or electrospun nanofiber membranes made from blended Poly Vinyl Alcohol materials with different degrees of hydrolyzation for optimal hydrogel degradation and sustained release of antiinfective silver salts" is an interesting article on the preparation of new Poly Vinyl Alcohol dressings by manufacturing electrospun membranes of Poly Vinyl Alcohol containing silver salts as anti-infective agent. The prepared material degrades in a controlled manner releasing silver in a sustained manner over 12 days. The topic of the article is interesting but is not well written and does not clearly report the obtained results. I recommend publication in the Nanomaterials after major revisions.

In the abstract the second sentence “In this new study we have further developed PVA dressings by manufacturing blended PVA films or electrospun membranes that contain silver salts and degrade in a controlled manner to release silver in a sustained manner over 12 days.” are unclear and should be re-written clearly describing the composition of new material. The “,” at the end of the sentence should be removed.

In the whole manuscript the word “anti-infective” is reported in different manner. Please uniform. In addition, in the different sections of the manuscript, the authors used the term “film” or “electrospun” simultaneously or separately, please select the most correct term and uniform.

In the result section, the Figures 1-4, 6-9 and the captions are not clear and exhaustive, for me. Please explain in the manuscript and caption the measured values and their significance and report what indicate the negative values. In my opinion, the results should be summarized and reported in a more rational manner.

In the section “Membrane electrospinning” of results, any results have been reported.

In the section “Silver release studies” of results, the authors reported the figure 5 after figure 9. In the caption of figure 5 and 10, the measured entities must be reported (like silver measured during …..) versus the time. In the figure, the legend “silver release” of “y” axis is not corrected.

Reviewer 3 Report

Abstract

The abstract has a very nonstandard structure. The structure should be – introduction, research gap, methodology, results, conclusion, outlook. The abstract should be rewritten to be more concise and more respective to the research done in the paper.

Introduction

The introduction has a quite weird structure, normally, it should start from general, going to specific. Also a research gap should be stated. The introduction should be revised and rewritten, it should cover the recent literature in the field, not to focus only on the previous author´s work. It seems that it is rather copied from the patent application, for which a different structure of outlining the problem/research gap can be present.

Please, check the spelling mistakes in the article, e.g. glutaldhyde instead of glutaraldehyde, but there are also others.

Methods

What is the origin of the bacterial strain (manufacturer), how was the “freezer stock” prepared?

The end of the method of antibacterial activity is unclear, please describe better/properly. How can you count CFU when you have liquid media/inocula?

You should check the English in the manuscript, mostly commas are missing, also space should be used between a value and a unit, etc. Many formal mistakes are present.

Results

Error bars in the graphs are totally missing, they must be added.

How many times were the experiments repeated and from how many replicates are the results shown?

Why was the degradation study done only for 6 h? Why some samples were not measured for the last time point? From the legends, it is absolutely unclear, what individual couloured lines are, even though some numbers are shown next to the plots, it is not properly described in the legends and thus it is unclear, what does it mean without reading the rest of the article. But figures should be understandable as standalone. In general, the legends should be improved.

Silver release studies – the methods should be mentioned both in the results description as well as in the methods. Again the figure legends in this part should be improved.

“Anti-biotic activity” – this should be renamed as “antibacterial activity”, the results are very, very poorly described, it must be significantly improved.

Discussion

Following references should be added after this sentence: “silver nanoparticles are now reported to be more effective and are regarded as the gold standard form of silver for wound dressings”

Silver Nanoparticles Stabilized Using Chitosan Films: Preparation, Properties and Antibacterial Activity (DOI: 10.1166/jnn.2015.11697)

Surface characterization and antibacterial response of silver nanowire arrays supported on laser-treated polyethylene naphthalate (DOI: 10.1016/j.msec.2016.11.072)

What is “Staph Aureus”? Be concise in writing terms. This is very vague.

Discussion of the antibacterial activity is very poor and comparing with other systems reported in the literature is totally missing.

Conclusion

Fully missing

References

Not in the right format, inconsistent, doi missing

Round 2

Reviewer 1 Report

The authors addressed all my questions, and the manuscript is recommended for publication now.

Reviewer 2 Report

Accepted in the present form

Reviewer 3 Report

The authors responded to all questions and significantly improved the manuscript as required. Therefore, I recommend the article for publication now.